# To Be or Not to Be Vaccinated? The Ethical Aspects of Influenza Vaccination among Healthcare Workers

**DOI:** 10.3390/ijerph16203981

**Published:** 2019-10-18

**Authors:** Wim Leo Celina Van Hooste, Micheline Bekaert

**Affiliations:** Occupational Health Services—External Service for Prevention and Protection at Work, Opvoedingstraat 143, B-9000 Gent, Belgium

**Keywords:** influenza, flu, vaccination, immunization, ethics, workplace, employees, healthcare workers, occupational medicine, mandate

## Abstract

Influenza is a highly contagious airborne disease with a significant morbidity and mortality burden. Seasonal influenza (SI) vaccination has been recommended for healthcare workers (HCWs) for many years. Despite many efforts to encourage HCWs to be immunized against influenza, vaccination uptake remains suboptimal. Sometimes there is a significant sign of improvement, only if numerous measures are taken. Is ‘the evidence’ and ‘rationale’ sufficient enough to support mandatory influenza vaccination policies? Most voluntary policies to increase vaccination rates among HCWs have not been very effective. How to close the gap between desired and current vaccination rates? Whether (semi)mandatory policies are justified is an ethical issue. By means of a MEDLINE search, we synthesized the most relevant publications to try to answer these questions. Neither the ‘clinical’ Hippocratic ethics (the Georgetown Mantra: autonomy, beneficence, non-maleficence, and justice), nor the ‘public health’ ethics frameworks resolve the question completely. Therefore, recently the ‘components of justice’ framework was added to the ethical debate. Most options to increase the uptake arouse little ethical controversy, except mandatory policies. The success of vaccination will largely depend upon the way the ethical challenges like professional duty and ethics (deontology), self-determination, vaccine hesitance, and refusal (‘conscientious objector’) are dealt with.

## 1. Introduction

Influenza is a highly contagious disease that causes a considerable burden of morbidity and mortality [1]. Flu is an acute viral infection that can lead to pneumonia, secondary bacterial infections, hospitalization and, occasionally, death, especially in certain risk groups. Healthcare workers (HCWs) who are generally healthy adults are not at high risk of serious complications following an influenza infection. However, they are a recommended target group for vaccination against seasonal influenza (SI) according to the World Health Organization [2]. Health professionals can be professionally exposed to the virus (aerogenic and contact with surfaces and objects) and can also act as a source of nosocomial infection of patients [1]. Account must always be taken of the fact that ‘healthcare workers’ refers to a diverse group and is therefore an umbrella concept. Unfortunately, there is a large heterogeneity in the studies on what is understood by a health professional (e.g., with or without direct patient contact) [1]. The major barriers to influenza vaccination have been well studied. They include a low skewed risk perception of influenza infection and also skepticism, incorrect beliefs and misconceptions about the safety and effectiveness of SI vaccines. The public, HCWs included, has to navigate through the post-truth and alternative facts era.

A meta-analysis of the incidence of influenza among healthcare workers and non-healthcare workers showed that healthcare workers have a significantly higher risk of influenza infection compared to employees outside this sector [3]. Although many scientists accept that flu vaccination benefits both the employee and the patient, there has been discussion about the extent of the benefit for patients and residents of residential care centers. A recent study by De Serres et al. demonstrated by extensive recalculations that the effect of the clinical studies, the clustered randomized controlled trials (cRCTs) on the benefit of patients and residents, is overestimated [4]. Nevertheless, since 1984 many agencies have recommended annual vaccination [5]. 

Our purpose is, by synthesizing the literature of the past decade, to make an overview of the relevant ethical issues arising around the occupational and public health topic ‘SI vaccination of HCWs’.

## 2. Methods 

We searched the MEDLINE database (2011 to 2019) to retrieve publications about the ethical and moral frameworks used to deal with SI vaccination among HCWs. The MeSH terms we used were: ‘influenza, human’, ‘vaccination’, ‘health personnel’, ‘ethics’, ‘morals’, ‘workplace’, and ‘occupational medicine’. The used free text terms were: ‘flu’, ‘immunization’, healthcare workers’, ‘mandate’, and ‘employees’. Further, we scanned the reference lists of all the full text papers to find more relevant publications. 

By using the three MeSH terms ‘influenza’, ‘vaccination’ and ‘ethics’, 136 publications were found. After abstract screening and combining with the free text term ‘healthcare workers’, 71 articles were found not to be relevant for healthcare workers and were excluded. Sixty-five publications were included for a full-test review. Five extra studies were identified through review of the reference lists of included studies.

## 3. Results

### 3.1. Vaccination Rate and Uptake 

The vaccination recommendations vary per country in terms of strength and target groups (risk persons such as pregnant women, health professionals, cohabitants of risk persons, etc.). The European Council recommends that Member States improve the vaccination rate in these target groups. A vaccination rate of 75% defined for risk populations can be used as a benchmark for health professionals [6]. Flu vaccination should be the rule rather than the exception [6]. A 75% benchmark in risk groups in the European Union (EU) is considered pragmatic and achievable [6]. Mandatory vaccination has proven to work in the United States of America to bring about more vaccinations, but it leads to conflict situations and causes disruptions in medical ethics for healthcare workers. Nevertheless, the vaccination rate of healthcare workers in the EU remains low up to suboptimal. However, is ‘empirical evidence’ sufficient to support a mandatory influenza vaccination policy in the EU to reach this 75% benchmark? As long as there is a proportion of employees who are not convinced of the benefits of vaccination, it will be difficult to achieve a predetermined goal of vaccination coverage [7,8]. In 2017, De Serres et al. calculated that 6000 to 32,000 health workers should receive the flu shot to prevent one death from influenza [4]. Some authors like Biondi et al. did wonder whether so much time, effort, and money should be spent on flu vaccination while other national healthcare priorities are put on the back burner [9]. 

### 3.2. Ethical Duty or Obligation? 

When accepting patients into their care, HCWs take a special professional fiduciary responsibility for their well-being, and that responsibility obligates us to follow all reasonable, evidence-based, best practices to ensure patients’ safety [10,11,12,13]. When a person chooses to work in healthcare, that person makes an autonomous choice to work in a service profession that serves the interests of vulnerable patients [11]. Certain obligations come with such choices. One is the obligation to take basic precautions to protect vulnerable patients against infections. It is not controversial that it is mandatory to wash hands or receive the hepatitis B vaccine. It must be uncontroversial that the simple, safe precaution of flu vaccination is also appropriate [11]. The vaccination would protect both the employee and the patients. If a health professional is not willing to take it, he or she will fail in his or her duty to patients [11]. Moreover, if an employee is unwilling to trust a medical intervention that is also supported by research, how can he trust the rest of the medical science that she apparently offers? [11].

Two widespread ethical principles of the Hippocratic medicine, namely ‘beneficence’ and ‘non-maleficence’, help to identify the ethical issues that arise when developing a position regarding the vaccination of healthy healthcare professionals. These two principles are therefore central to medical ethics: (1) promoting the well-being of the patient, and (2) not harming the well-being of the patient [12]. The ethically justified conditions for mandatory flu vaccination according to Wicker et al. are that (1) flu vaccinations must be effective to prevent influenza infections in general and in particular in healthcare workers and that (2) there must be sufficient empirical evidence that flu vaccination of health professionals reduce mortality and morbidity in patients and residents [12]. If these two conditions are met, there are good ethical reasons to promote, possibly oblige, flu vaccination among healthcare workers. These professionals have a special ethical obligation not to harm their patients or residents (the principle of ‘primum non nocere’—first of all, do no harm) (‘harm principle’) [11,12]. Caplan argued for mandatory vaccination because: (1) every ethical code used by doctors, nurses, and other healthcare professionals states very clearly and concisely that patients’ interests must take precedence over those of others; (2) health professionals are obliged to respect the fundamental medical ethical requirement of ‘primum non nocere’, and lastly (3) HCWs have a special role to play with regard to vulnerable individuals who cannot protect themselves [14]. This is a duty that is generally recognized in professional ethical codes. By not vaccinating themselves, they feed the fear of vaccines, reinforce the anti-vaccination feelings and set a bad example for the public [14]. The ethical obligation to be vaccinated rests with each individual employee as an extension of their professional duty to their patients. Further, health care institutions also bear an ethical obligation to ensure, maintain, and enforce universal SI vaccination among their staff, given their ability to implement vaccination policies and their increasing individual agency and institutional fiduciary responsibilities in the modern medical era [10,11,12,13]. Policies of mandatory vaccination against influenza are in line with professional ethics. This obligation benefits many, including some who have to rely on health professionals to protect them. Furthermore, it is good to maintain a stable workforce. An example is set that allows for fair involvement with others working in the hospital, as well as with the general public to take the right position on vaccination [14].

### 3.3. Three Ethical Models

Three models are used to address the ethical issue of flu vaccination among health professionals: (1) the Hippocratic model, (2) the ‘New Ethics’ model, and finally the most recently model, (3) ‘Components of justice’ model [15]. Given the historical emphasis within Hippocratic medicine on ‘primum non nocere’, the risk that the health professional poses to patients and colleagues is an important point of discussion. Given the risk of acting as a vector prior to the symptomatic period and the chance of transmission by air, vaccination is the most viable option to interrupt the transmission of the influenza virus. Professional licenses for practicing medicine, nursing, or related fields result in special privileges and responsibilities. Vaccination can be seen as one of those responsibilities [11,12,16]. In the ethics model based on Hippocrates and the Georgetown Mantra based on the four ethic Hippocratic principles (autonomy, beneficence, non-maleficence, and justice), the doctor works for the well-being of the individual patient and the voluntary nature of the relationship between patient and doctor is axiomatic [17]. The rule utilitarian ‘New Ethics’ model, however, attempts to replace the clinical Hippocratic model with a public health model. The emphasis is on prevention and on optimizing public health, not on individual outcomes. An expert committee replaces the individual patient as a decision maker [17]. Mandatory vaccination is the leading factor in the new ethics point of view. The policy of requiring annual flu vaccination as a condition for working in a medical facility illustrates the dogmatism of the public health model and how it transcends individual autonomy, Hippocratic ethics, and also evidence-based medicine [17]. Orient argues that the strength of the evidence appears to be inversely proportional to the moral zeal of the mandatory proponents of vaccination, referring to Caplan’s publication from 2011 [14,17]. The mandatory influenza vaccination is then based on authority and not on facts [17].

The clinical and public health perspective frameworks have led to arguments for and against making vaccination mandatory. On the one hand, the clinical ethical frameworks support the mandatory immunization based on the duty of health professionals to improve the well-being of patients. On the other hand, violation of the employee’s autonomy is mentioned as a primary objection to the requirement of immunization. The public health ethics frameworks have been used to justify mandatory programs by supporting the claim that the protection of the community through immunization is fundamental to public health [18]. However, public health ethics frameworks have also been used to support the claim that there is no direct evidence that mandatory vaccination programs prevent disease in patients and should therefore not be implemented [19]. Neither the clinical nor the public health ethics frameworks solve the question about mandatory SI vaccination policies ‘completely’. Lee added justice components to provide a more comprehensive defense for requiring influenza immunization [15]. Lee starts with the fundamental moral principle that we have a duty to protect the vulnerable. Although this duty also has moral weight in daily life, it is especially prominent in the healthcare worker-patient relationship. HCWs have a duty to implement a comprehensive approach to ensure that medically vulnerable patients are protected [15].

However, Galanakis et al. divided the ethical issue into three other sections: (1) the professional ethics of the employee, (2) the institution-specific ethics, and (3) ethics according to public health [12]. The clinical ethics of the health professional state that vaccination is consistent with a collective professional obligation, and being immune is a responsibility of the employee. Health institutions have a moral obligation to reduce the risk of infection of patients and residents, as well as to protect their employees and keep them available during flu epidemics. Preventing the spread of infectious diseases is a top priority in public health. The right of the community to be protected against infectious diseases is more important than the freedom of health professionals to refuse the vaccine [12].

In any case, mandatory vaccination policies should be based on a transparent and fair decision-making process [12,13]. Mandatory vaccination has proven to work and accomplish more vaccination but causes disturbance in medical ethics for HCWs. Many HCWs claim that it is unethical to mandate vaccination when it is inconvenient, lacks evidence, is ineffective, and is potentially risky [20]. The infringement on HCW autonomy has been cited as a primary objection to requiring immunization [21].

After decades of vaccine use, it is hard to detect any public health impact. This is in strong contrast to other routine vaccinations, such as polio and *Haemophilus influenzae type b*, where the introduction of the vaccine led to obvious decline of the disease (9). The influenza vaccine has been shown to only moderately effective for healthy adults (about 60% to 70 %), so the SI vaccination does not guarantee immunity against influenza for HCWs. HCWs are not the only source of influenza for hospitalized patients. There are no good quality studies suggesting vaccinating HCWs against influenza protects patients in hospitals from laboratory-confirmed influenza [22]. Kelly argued that: (1) healthcare workers can be considered to be healthy working adults, a group for which the influenza vaccine has proven to be moderately effective, influenza vaccination is not a guarantee of immunity to influenza for healthcare workers; (2) health workers are not the only source of influenza for hospitalized patients; and (3) that there are no studies of good quality to indicate that vaccination of workers against influenza protects patients in hospitals against laboratory-confirmed influenza [22]. 

The ‘omission bias’ refers to the belief that causing harm through inaction is more acceptable than causing harm from action, and this bias possibly operates in HCWs who decline influenza vaccination [23]. Ethical principles of autonomy, non-maleficence, and altruism together with the professional and legal framework are needed to be considered before implementation of a mandatory policy [23]. In an effort to enhance vaccination coverage, there is the need to build stronger and more extensive scientific evidence for supporting the development of practice guidelines [23]. 

### 3.4. The Empirical Evidence

The recurring question is whether the evidence of the efficacy of the flu vaccination provided today is inadequate. In 2012 Abramson studied four randomized studies forming the basis for the Society for Healthcare Epidemiology in America (SHEA) to recommend mandatory vaccination [24]. Abramson came to this conclusion: “the repeated conclusion that vaccination of personnel has preventive value for elderly patients in nursing homes seems to be the result of major methodological errors and wishful thinking”. According to Abramson, the arguments for influenza vaccination are not supported by the published literature [24]. There are several studies that have analyzed the effectiveness of SI vaccination by health professionals and the benefits for patients and residents. A systematic review by Thomas et al. in the Cochrane Database of Systematic Reviews of pooled data from three cRCTs showed lower all-cause mortality and non-specific morbidity among elderly residents of long-term care but did not show that influenza vaccination reduced serologically proven influenza in people aged 60 and over [25]. An update of the Cochrane Database of Systematic Reviews was published in 2013, which concluded, “there is no reasonable evidence to support the vaccination of health professionals to prevent influenza in people aged 60 and over in long-term care institutions” [26]. An update of the Cochrane systematic review was published in 2016, confirming the analyzes from 2010 and 2013 [27]. The evidence from observational studies has proven that mandatory vaccination increases the vaccination rate, but there is no evidence of clinical outcomes [28,29]. Further studies documenting the impact of healthcare worker influenza vaccination on clinical outcomes could influence decisions on the use of mandatory vaccine policies in the health sector [28,29]. Prospective studies across multiple facilities are likely to be necessary to obtain sufficient evidence to evaluate healthcare professionals and clinical outcomes of patients after flu vaccination in acute care settings [30]. Assessing clinical outcomes will be a challenge, but major healthcare employers who are planning to implement a mandate need to develop a strategy to evaluate those outcomes and patient health outcomes. 

To increase the vaccination rate, there is a need for stronger and more comprehensive scientific evidence to support the development of practical guidelines [30]. On the other hand, Sundaram et al. postulate that more data is not necessarily needed, but more transparency, including a professional synthesis based on knowledge and uncertainties about the quality of the ‘evidence’, together with an explanation why the flu vaccination should nevertheless be recommended [31].

In 2017, De Serres et al. concluded that an “intuitive feeling” that there is some evidence of any patient benefit is an insufficient scientific basis for ethically abolishing the individual rights of health professionals [4]. They also argued that those who are in favor of mandatory vaccination have a duty to provide reliable evidence, certainly in the absence of good studies demonstrating influence on patient safety [4]. The absence of these studies does not prevent the support of voluntary vaccination campaigns, as well as the other protective measures (staying at home in the event of illness, mask, a proper hand hygiene, etc.) [4].

Table 1 shows a synthesis of the publications from the 2011–2019 period regarding some arguments in favor of or against (mandatory) SI vaccination of HCWs (Table 1).

## 4. Discussion

It is unlikely that purely voluntary programs will achieve vaccination rates that are sufficient to meet the benchmark of 75% uptake among HCWs. The generally accepted ethical principles of professional and fiduciary duty, beneficence and non-maleficence (primum non nocere), were often found in the retrieved publications. The ethical arguments pro and contra (semi)mandatory SI vaccination policies do not solve the gap between desired and current vaccination rates. Most options to increase the uptake arouse little controversy, except mandatory policies. Mandatory vaccination policies cause, e.g., disturbances in the relationship with the employer and have led to legal entanglements. 

One of the weaknesses of this publication, is the use of solely the MEDLINE database to retrieve articles. Another weakness could be the exclusion of non-influenza vaccination ethics. But the SI vaccination is a special one among the immunizations because an annually application is necessary. Therefore, our scope was fixed on the ethics of yearly flu vaccination. Further, this paper is neither a systematic review nor a meta-analysis because we think it is difficult to perform such thing for ethical issues.

When we compare our overview of publications from the 2011–2019 era with the 2000–2010 period, we found 52 papers of interest. Out of these earlier papers, we selected 13 papers of interest for full-text review [32,33,34,35,36,37,38,39,40,41,42,43,44]. This paper retrieved more articles than in the previous years, but we found the same highlights in the ethical debate as the decade before. For instance, putting patients and professional ethics over personal preference [33]; effectiveness, beneficence, necessity, autonomy, justice, and transparency [39]; and the two responsibilities of HCWs, namely, responsibility as health professional and as member of the collective [44]. Moreover, pro/contra discussions were previously published [35,36,40,41]. Van Delden et al. point out that “when uptake falls short a mandatory programme may be justified” [38]. Poland et al. made this remark: “high rates of HCWs immunization will benefit patients, HCWs, their families, and employers, and the communities within they work and live” [42].

## 5. Conclusions

Despite many years of great efforts and the widespread recommendation around the world, the SI vaccination rate among HCWs seldom exceeds a proposed 75% benchmark. Mandatory policies in certain healthcare facilities have reached this goal. Especially when taking care of the most vulnerable, there are arguments to include a form of vaccination obligation (harm principle). Despite the ongoing debate about the evidence in systematic meta-analyses, several healthcare organizations have found that the evidence is strong enough to justify a mandatory vaccination policy for healthcare workers. However, it remains controversial to make SI vaccination mandatory, in particular due to ethical and legal implications. The different ethical approaches, principles, concepts, models, and frameworks do not answer all questions. There is always a tension, i.e., conflict, between ‘primum non nocere’ (Hippocrates), ‘beneficence’, ‘non-maleficence’, professional obligation, and duty (deontology); personal autonomy and individual freedom; vaccine hesitance and refusal (‘conscientious objector’). This makes it a challenging but also open issue: “To be or not to be vaccinated? That’s the question for healthcare workers!”.

The Main Conclusions:Seasonal influenza vaccination uptake among healthcare workers remains low on to suboptimal around the world, especially when a 75% benchmark is taken into consideration.Serious and complex ethical hurdles to increase healthcare workers vaccination rates must be taken. For example, subsuming the individual freedom for collective and institutional goals to obtain herd immunity, a ‘public good’, and protect the most vulnerable patients and residents. There is a very little downside to vaccination (a tiny risk of severe side-effects) to do a world of ‘good’.There are many individual and collective duties for healthcare workers: a duty to patients (communitarian altruism), a duty to protect oneself, duty to one’s family, duty to colleagues and duty to society (solidarity).

## Figures and Tables

**Table 1 ijerph-16-03981-t001:** Synthesis of publications from the 2011–2019 period regarding some arguments in favor of or against (mandatory) Seasonal Influenza (SI) vaccination of Healthcare Workers (HCWs).

Year	Author(s)	Reference	Arguments Pro/Contra	Mandate for Vaccination	Remarks
2011	Caplan	[14]	“A duty” for HCWs, “Set an example” as HCWs, “To do the right thing”	Yes	“New ethics” point of view
2012	Abramson	[24]	“major methodological errors and wishful thinking”	No	
Orient	[17]	“the SI vaccine mandate is authority-based, not evidence-based”	No	
2013	Galanakis et al.	[12]	“professional obligation” of the HCWs, “moral obligation” of the health institution, and “preventing infectious diseases is top priority in public health”	Yes	
Thomas et al. (Cochrane Database Systematic Review)	[26]	“no reasonable evidence to support the SI vaccination of HCWs”	No	
2014	Born	[29]	“evidence linking HCW vaccination to patient outcomes in the absence of a mandate is limited”	No	
Cortes-Penfield	[10]	“professional fiduciary responsibility for well-being of patients”	Yes	Healthcare institution bears an ethical obligation to ensure and enforce universal SI vaccination among their staff
Pitts et al.	[28]	“prospective studies across multiple facilities would likely be needed to obtain sufficient power to evaluate HCW clinical outcomes”	-	Further studies on clinical outcomes are necessary
Wicker & Marckmann	[11]	“2 conditions for mandatory SI vaccination must be full-filled: (1) SI vaccination must be effective, and (2) sufficient empirical evidence”	-	In any case, mandatory vaccination policies should be based on a transparent and fair decision-making process
2015	Biondi et al.	[9]	“should so much time, effort, and money be dedicated to flu vaccination while other healthcare priorities remain on the back burner”	No	“healthy user bias”
Dubov et al.	[20]	“mandatory vaccination causes disturbances in medical ethics of HCWs”	No	unaware, unbelieving, unmotivated, and unconcerned HCWs
Kelly	[22]	“HCWs can be considered as healthy working adults”“HCWs are not the only source of influenza for patients”“there are no good quality studies”	No	
Lee	[15]	“neither the clinical nor the public health ethics frameworks resolve the question ‘fully’”“HCWs have an obligation to protect the vulnerable”“’this moral is especially salient in the healthcare relationship”“an obligation to implement a comprehensive approach to ensure that medically vulnerable patients are protected”	Yes	Components of justice
2016	Najera et al.	[13]	“when a person chooses to go into healthcare, that person makes an autonomous choice to work in a service profession, serving the interests of vulnerable patients”“with such choices come certain obligations”	Yes	
To et al.	[23]	“ethical principles of autonomy, non-maleficence, and altruism together with the professional and legal framework are needed to be considered before implementation of a mandatory policy”“there is a need to build stronger and more extensive scientific evidence for supporting the development of practice guidelines”	No	“omission bias”
2017	DeSerres et al.	[4]	“the effect of the clinical studies, the CRCTs, on the benefit of patients and residents is overestimated”	No

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
