# Peer review of "To Be or Not to Be Vaccinated? The Ethical Aspects of Influenza Vaccination among Healthcare Workers"

_ijerph, 2019, doi:10.3390/ijerph16203981_

Round 1

Reviewer 1 Report

Dear Authors,

I read with interest your review on the ethical aspects of influenza vaccination among healthcare workers.

In my opionion could be considered for publication in IJERPH.

Some minor revisions should be considered.

Results Section: when you discuss the influenza vaccination of HCWs as a "Ethical duty or obligation?" you could add a consideration about the possible previous transmission of influenza viruses to HCWs and later to patients (as suggested by "Restivo V, et al. Influenza like Illness among Medical Residents Anticipates Influenza Diffusion in General Population: Data from a National Survey among Italian Medical Residents. PLoS One. 2016;11(12):e0168546. doi: 10.1371/journal.pone.0168546."

Moreover, in the discussion section could be discussed the issues recently raised by "Costantino C, at al. Influenza vaccination of healthcare workers in Italy: could mandatory vaccination be a solution to protect patients?. Future Microbiol. 2019 Jun 18;. doi: 10.2217/fmb-2018-0238." that analyzed the limited impact of communicative and formative campaigns on HCWs and the need of a mandatory influenza vaccination strategy.

Author Response

Reviewer 1:

Improved the research design & described the methods

1)

Abstract: A Paragraph was added

Included a sentence about the methods and the objective.

2)

Introduction

Purpose of the article was added 

Included sentence of the purpose

3) Methodology

We described the applied methods in more detail  

4) Discussion

Included references from the 2000-2010 ear and made a comparison with these studies and added strength and weaknesses, the limits of the study

Reviewer 2 Report

This study is important and it puts a light on MEDLINE database 2011 to 2019 about the ethical and moral framework to deal with health care workers. The main conclusions are highlighted clearly. 

The methods could be improved, by adding more data. Data could be more evidently presented by graphs or tables. 

Author Response

Reviewer 2:

Revision of the description of the research design & methods

1)

Describing the applied methods in more detail

2)

Presentation is presented in Table 1

Reviewer 3 Report

The article is interesting especially in the current context of questioning the usefulness of vaccinations.

Nevertheless, I wonder if this article should not target a journal of medical ethics rather than a public health journal.

Several points can be improved for publication:

The article is constructed and written as if it were a book chapter and not really as a scientific article. The subject is very much based on the same article by De Serres. His article is interesting but basing your point of view mainly on a single article is a little weak in terms of evidence.

. Abstract :

The Abstract is well constructed, but there is nevertheless a paragraph missing on the methods used by the authors and the objective of the study

Introduction.

The introduction is well documented, but the clearly stated purpose of this article is missing at the end of this chapter.

Methodology

This part should be further developed. The methods used to analyze the articles and remove key messages are missing. A flow chart on the number of articles found and then selected and analyzed with the N would be useful

Discussions

The discussion seems to me to be incomplete, it has no reference, no comparison with other studies (validated externally and internally) and no limit to the study.

Author Response

Included Table 1:

Table 1: Synthesis of publications from the 2011-2019 period regarding some arguments in favor of or against (mandatory) Seasonal Influenza (SI) vaccination of Healthcare Workers (HCWs)

Round 2

Reviewer 3 Report

I am satisfied with the authors' additions and corrections, and I think that this article can be accepted for publication in this form.